# Assessment of Influencing Factors on the Spatial Variability of SOM in the Red Beds of the Nanxiong Basin of China, Using GIS and Geo-Statistical Methods

Ping Yan [1], Kairong Lin [1,2,3,*], Yiren Wang [1], Xinjun Tu [1], Chunmei Bai [1] and Luobin Yan [4]

1 School of Civil Engineering, Sun Yat-sen University, Guangzhou 510275, China; yanp8@mail.sysu.edu.cn (P.Y.); wangyr55@mail.sysu.edu.cn (Y.W.); eestxj@mail.sysu.edu.cn (X.T.); baichm@mail.sysu.edu.cn (C.B.)
2 Guangdong Key Laboratory of Oceanic Civil Engineering, Guangzhou 510275, China
3 Southern Marine Science and Engineering Guangdong Laboratory, Zhuhai 519000, China
4 School of Geographical Sciences, Southwest University, Chongqing 400715, China; yanluobin@swu.edu.cn
* Correspondence: linkr@mail.sysu.edu.cn

Kostas Kalabokidis and
Wolfgang Kainz

**Abstract:** Understanding the spatial variability of soil organic matter (SOM) is crucial for implementing precise land degradation control and fertilization to improve crop productivity. Studying spatial variability provides a scientific basis for precision fertilization and land degradation control. In this study, geostatistics and classical statistical methods were used to analyze the spatial variability of SOM and its influencing factors under various degrees of land degradation in the red bed area of southern China. The results demonstrate a declining trend for SOM content with increasing land degradation. The SOM content differs profoundly under different land degradation degrees. The coefficient of variation ranges from 13.61% for extreme land degradation to 8.98% for mild land degradation, 7.96% for moderate land degradation, and 5.64% for severe land degradation. A significant positive correlation is displayed between the altitude and the SOM ($p < 0.01$) under mild and moderate land degradation conditions. Bulk density and pH value have a significant negative correlation with SOM ($p < 0.01$). It can be observed that terrain factors, as well as physical and chemical soil parameters, have a great influence on SOM.

**Keywords:** soil organic matter; land degradation; semivariogram; spatial distribution; classical statistics

## 1. Introduction

Soil organic matter (SOM) is a primary source of plant mineral nutrition and an essential part of the terrestrial soil carbon pool. The content and dynamics of SOM directly impact the global carbon cycle [1–3]. Land degradation is affected by intersecting factors including topography, parent material, climate and human activities. Soil nutrition loss is associated with the condition of land degradation, which affects the health of the soil carbon pool [4,5]. Therefore, it is important to carry out reasonable soil and water conservation measures to promote soil carbon recovery and accumulation [6]. Identifying the influencing factors of SOM variability under different land degradation levels establishes a theoretical basis for studying regional soil carbon restoration mechanisms, thereby facilitating the development of soil quality restoration, ecological reconstruction, and water and soil conservation in ecologically fragile areas. The conservation, migration and distribution of SOM are complex physical, chemical and biological processes affected by many factors. The past literature has shown that because of the variation in soil erosion, the SOM content varies in different regions [7–10], land-use patterns [11,12] and landforms [13,14]. For example, Yao et al. [15] studied the change in SOM content in different soil depths (0–20, 20–40 and 40–60 cm) in the red soil region of South China, and found that the spatial pattern of SOM was characterized by higher content in the periphery and lower content in the middle. Zhang et al. [14] analyzed the spatial heterogeneity of SOM in the Karst mountain

area which possesses a fragile ecology. Their statistical results indicate that the landforms, which lead to great discrepancies in human activities and geographic characteristics, are the primary factor for the high heterogeneity of SOM content in mountainous Karst areas. Thus, investigating the influencing factors for the spatial variability of SOM under different land degradation degrees contributes to improving mappings of SOM spatial variability and estimations of the recovery potential for the soil carbon pool in ecologically vulnerable areas.

The Nanxiong basin is a typical red bed ecosystem with special material, energy, structure and function, which is formed based on red sandstone and glutenite and under the interaction of the atmosphere, water, rocks and organisms [16,17]. In recent years, the aggravation of human activities has exacerbated the land degradation of red bed ecology, generating severe ecological problems such as deserts in the red bed area of southern China [18]. The loss of SOM has become an obstacle to the sustainable development of local agriculture, and the protection of soil environment and land resources. Some publications [19–23] have illustrated that the variation of SOM content is associated with terrain factors (altitude, slope), bulk density, soil pH, etc. However, the research into the influencing factors of SOM spatial variation, focusing on different land degradation degrees, is still lacking and has become a pressing topic. Thus, it is of great significance to explore the spatial variability of soil organic matter and its associated impact mechanism under various land degradation types in the red bed area. As a result, this research uses the ecologically fragile area of southern China as a study area, and investigates the relationship between the SOM content and topographic factors, soil physical and chemical parameters in different land degradation conditions, providing a foundation for farmland utilization, and conservation of soil and water.

Therefore, the main objectives of this study are: (1) to clarify the spatial variation characteristics of SOM under different land degradation types in this study area; (2) to obtain the influencing factors of spatial variation of SOM through correlation analysis; and (3) to explore the impact mechanism of land degradation on the spatial distribution pattern of SOM in the study area.

## 2. Materials and Methods

### 2.1. Description of the Study Area

The study area is located in the northeast of Nanxiong basin, a typical red bed area in Nanxiong City, Guangdong Province (Figure 1). Dayuling is located in the north, surrounded by mountains on three sides, including Jiangxi province in the east, Renhua in the west and Shixing in the south. The geographical coordinates range from 114°29′50″ E to 114°33′30″ E, and from 25°13′23″ N to 25°16′57″ N. It consists of a series of purple-red mudstone, siltstone, silty conglomerate, red-gray glutenite, granitic conglomerate and glutenite. Shallow hills with an altitude of 105–250 m are distributed widely in this area. The area has a subtropical monsoon humid climate with four distinct seasons throughout the year. The annual average temperature is 19.6 °C, the annual average evaporation is 1678.7 mm, and the annual average rainfall is 1555.1 mm. The natural soil types are yellow soil, red soil, red lime soil and purple soil.

### 2.2. Field Sampling and Laboratory Testing

The sample was collected in November 2017 after the local crops were fully harvested. A global positioning system (GPS) was used to determine the longitude, latitude, and altitude and other information of each sampling point. A geological compass was employed to obtain the slope and rock strike of sampling points. At a spacing of around 50–300 m, a 5 cm-diameter hand-held soil drill was used to collect soil samples at a depth of 0–20 cm, after removing the debris and growing plants on the sampling surface. Five to six samples were collected from each sampling location and mixed evenly using the quartering method to obtain a 1 kg combined soil sample, which was put into labelled plastic bags.

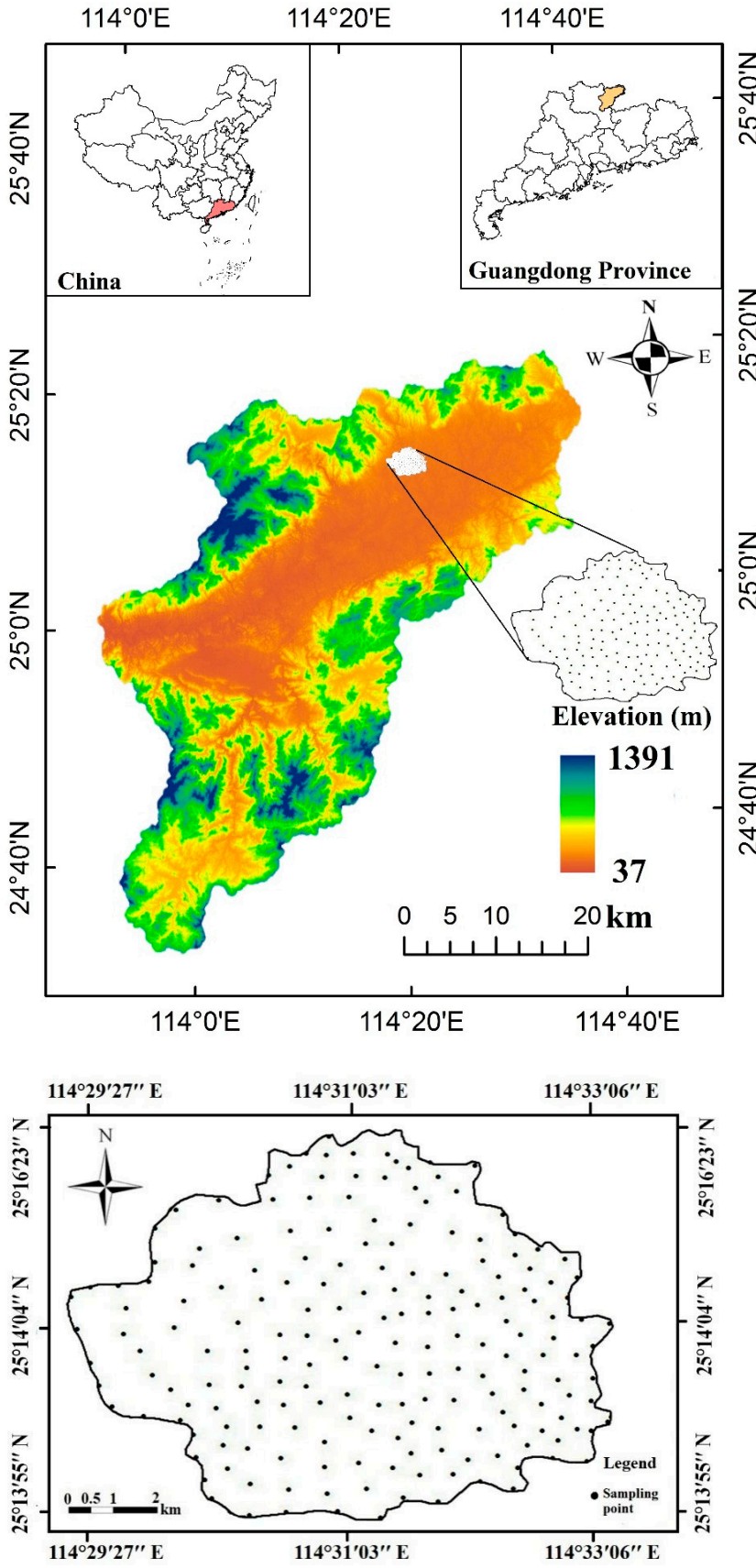

**Figure 1.** Location of the study area and distribution of soil sampling sites.

### 2.3. Laboratory Testing

In the laboratory, the soil samples were dried, and weeds and impurities were removed using a 2 mm sieve. The method of sodium dichromate wet oxidation was used [24,25]. The principle of this method is to oxidize the soil organic matter with an excess of standard potassium bicadmium sulfuric acid solution. The remaining potassium bicadmate was titrated with ferrous sulfate solution, and the amount of potassium bicadmium consumed was used to calculate the soil organic matter content. The chemical reaction formula of the determination process is as follows:

$$2KCr_2O_7 + 3C + 8H_2SO_4 + 2K_2SO_4 \rightarrow 2Cr_2(SO_4)_3 + 3CO_2 + 8H_2O \tag{1}$$

$$KCr_2O_7 + 6FeSO_4 + 7H_2SO_4 \rightarrow K_2SO_4 + Cr_2(SO_4)_3 + 3Fe_2(SO_4)_3 + 7H_2O \tag{2}$$

Specific operation steps were as follows: (1) 0.500 g of the air-dried soil sample passing through 2 mm sieve was weighted and transferred into a 150 mL Erlenmeyer flask; 5.00 mL of 0.800 mol/L $K_2Cr_2O_7$ solution and 5 mL concentrated $H_2SO_4$ were added using a pipette; the solutions were mixed evenly by shaking and covered with a bent-neck funnel to condense and vaporize; (2) the solution was heated on a preheated electric sand bath with a temperature of 170–180 °C; we started timing when it began to boil to allow boiling to last for $5 \pm 0.5$ min; (3) the triangular flask was then taken off and cooled, then the solution was carefully transferred into a 250 mL triangular flask, and the small funnel and original flask were rinsed with distilled water, which was then poured into the 250 mL; note that the total solution volume was controlled within the range of 60–80 mL; (4) 3 drops of o-phenanthroline indicator were added and titrated with 0.2 mol/L $FeSO_4$; the solution color first changed from orange-yellow to blue-green, and then changed to brick-red at the end of the titration; the amount of $FeSO_4$ was recorded as $V$; this step was repeated three times; and (5) while testing the sample, two blank tests were conducted and the average value was calculated; burned soil or quartz sand were used to replace the soil sample to avoid spilling the solution; the same procedures were followed and the used $FeSO_4$ amount was recorded as $Vo$. The calculation formula is as follows:

$$\text{SOM}(\%) = \frac{(V_O - V) \times N \times 0.003 \times 1.724 - 1.1}{w} \times 100\% \tag{3}$$

where N is the molar concentration of ferrous sulfate; 0.003 is the 1/4 carbon atom millimolar mass; $w$ is the dried soil weight, and 1.724 is the conversion coefficient between organic carbon and organic matter.

All laboratory experiments were completed by the Key Laboratory of Comprehensive Management of Agricultural Environment of Guangdong Province, Institute of Eco-environmental and Soil Science, Guangdong Academy of Sciences, China.

### 2.4. Data Analysis

Classical statistics and geostatistics methods were used for data analysis. The spatial distribution map for SOM was drawn using the Kriging interpolation function within the geostatistics module of the ArcGIS 10.5 software. Based on the different sampling densities and soil type, the random grid was set up under four land degradation types (i.e., mild, moderate, severe and extreme) for a total of 225 sample points. The spatial variability and correlation of soil organic matter were analyzed using the GS + 9.0 and ArcGIS 10.5 software [26,27]. The calculation formula of the semivariogram can be presented as follows:

$$\gamma(h) = \frac{1}{2N(h)} \sum_{i=1}^{N(h)} [Z(x_i) - Z(x_i + h)]^2 \tag{4}$$

where $\gamma(h)$ is the number of points with a distance of $h$, $Z(xi)$ is the sample value at position $xi$, $Z(xi + h)$ is the value at distance $x_i + h$, and $N(h)$ is the number of samples separated by $h$.

### 2.5. Classification of Land Degradation Degree in Red Bed Area

Based on our wide range of field investigations, land degradation in the study area is categorized into four levels: mild, moderate, severe and extreme land degradation. Mild land degradation has a certain amount of soil coverage above the majority of red bed soft rock, and a coverage rate of 50–70%, including sparse forest or grass shrubs, but its dominant native vegetation has gradually reduced, leading to the appearance of red bed bare areas with spots. Moderate land degradation is observed to have a vegetation coverage rate of only 30–50%, with the occurrence of patchy bare rock in strong erosion areas, the formation of a large number of small erosion ditches, the rapid decline of community biomass, and strongly eroded surface soil. Severe land degradation possesses a coverage rate of only about 10–30%, including sparse forest or shrub grass cover, with flaky red soft rock exposure, heavy gully erosion, local patchy soil layers, and the growing of some dry soil arid shrub and grass vegetation. Extreme land degradation has a vegetation coverage rate of less than 10%, with more than 90% of the surface consisting of red bed soft rock outcrops, dense surface erosion ditches, no mature soil on the surface, and clusters of drought-tolerant herbs or shrubs. According to the above categorization descriptions, the specific quantitative characterization values of land degradation degrees are shown in Table 1. The Land Degradation Classification System was developed by the plot investigation in the Nanxiong Basin, conducted by one of the authors in this study, Luobin Yan.

**Table 1.** Classification of red bed land degradation at landscape scale.

| Grading Standard | Naked Features | Soil Characteristics | Vegetation Characteristics | Land Production Potential |
|---|---|---|---|---|
| Mild land degradation | Spotty bedrock exposed | Most of the soil layers are more than 50 cm thick, with complete ABC soil layer and slight soil erosion. | About 50–70% vegetation coverage, and the community structure is complex, forming an obvious interlayer structure of arbor, shrub and grass. | Biological production capacity is high, and can be used for forestry or agricultural land. |
| Moderate land degradation | Patchy bare rock outcropping | Most of the soil layer is 20–50 cm thick, only BC layer, humus (A) development is not obvious, soil erosion is strong. | 30–50% vegetation coverage, the arbor layer is destroyed to form shrub grass communiteis, with artificially planted Pinus massoniana and Schima superba forests. | The potential productivity of land is relatively low, and can be developed as irrigated land, dry land or artificial economic forest land. |
| Severe land degradation | Exposure of flaky bare rock | Most of the soil layer is 5–20 cm thick, with thin eluvial layer (B), and the soil erosion is severe. | 10–30% vegetation coverage, and the community is dominated by grass slope meadow with few plant species and interspersed with drought tolerant thorny shrubs. | The potential productivity of land is scant, can mainly be used for uncultivated dry land, artificial eucalyptus, leucaena shelter forest land and so on. |
| Extreme land degradation | Continuous bedrock exposure | The thickness of the soil layer is less than 5 cm, with only weathered debris. The process of soil formation is not obvious, the loss is rapid, and the weathering erosion of the bedrock is strong. | Less than 10% vegetation coverage, with only a few extremely drought-tolerant shrubs and herbs distributed. | There is basically no biological production potential. |

## 3. Results

### 3.1. Descriptive Statistical Characteristics of SOM

According to the statistical analysis results of SOM (Table 2), the average value of SOM is 19.84 g/kg, ranging from approximately 9.11 to 34.80 g/kg. With the exacerbation of land degradation, the SOM content showed a downward trend, with mild land degradation (27.70 g/kg) > moderate land degradation (21.11 g/kg) > severe land degradation (17.02 g/kg) > extreme land degradation (13.45 g/kg).

K-S test results showed that the soil moisture in the study area follows a normal distribution ($p > 0.05$) and met the requirements of geostatistical analysis. It can be concluded that all of them conform to normal distribution. The coefficient of variation of SOM

under different types of land degradation was at a medium level. The order of coefficient of variation was extreme land degradation (13.61%) > mild land degradation (8.98%) > moderate land degradation (7.96%) > severe land degradation (5.64%).

**Table 2.** Descriptive statistic characteristics of SOM in soil with different land types.

| Types of Land Degradation | Samples | Minimum | Maximum | Average | Standard Deviation | Coefficient of Variation | Skewness | K-S L-Test |
|---|---|---|---|---|---|---|---|---|
| Total | 225 | 9.11 | 34.80 | 19.84 | 2.64 | 13.31 | 0.42 | 1.65 |
| Mild land degradation | 57 | 24.25 | 34.80 | 27.70 | 2.49 | 8.98 | 0.56 | 1.66 |
| Moderate land degradation | 56 | 18.36 | 24.21 | 21.11 | 1.68 | 7.96 | 0.36 | 1.61 |
| Severe land degradation | 55 | 15.42 | 18.36 | 17.02 | 0.96 | 5.64 | −1.51 | 0.93 |
| Extreme land degradation | 57 | 9.11 | 15.39 | 13.45 | 1.83 | 13.61 | −0.78 | 0.62 |

### 3.2. Semivariogram Analysis

It can be seen from Table 3 that the nugget effect value (i.e., the proportion of spatial variation caused by randomness in the total variation of the system) of SOM under different land degradation types is less than 25%, indicating that there is a strong spatial correlation and that the internal factors play a major role. The impact of terrain factors and soil structural factors on the spatial variation of SOM reaches 89%. The semivariogram model of SOM under different land degradation types follows Gaussian distribution. The SOM level ranged from 2646.57 to 2824.97 m. The sampling design scale of this experiment was 50–300 m, which met the requirements of geostatistics sampling and reflected the spatial pattern information of SOM. Under different land degradation types, the range from large to small was mild land degradation (2824.97) > moderate land degradation (2805.92) > severe land degradation (2769.55) > extreme land degradation (2646.57), indicating that the spatial autocorrelation distance of mild land degradation was large. The second is moderate land degradation, and the autocorrelation distance of extreme land degradation is small. The order of spatial autocorrelation intensity was mild land degradation (10.56%) > moderate land degradation (7.85%) > severe land degradation (7.76%) > extreme land degradation (0.33%). With the increase in land degradation level, the spatial autocorrelation intensity tends to decrease.

**Table 3.** Semi-variogram parameters of SOM in soil with different vegetation types.

| Types of Land Degradation | Number of Samples | Model | Nugget | Sill | Nugget/Sill | Rang (m) | $R^2$ |
|---|---|---|---|---|---|---|---|
| Mild land degradation | 57 | Gaussian | 0.15 | 1.42 | 10.56 | 2824.97 | 0.98 |
| Moderate land degradation | 56 | Gaussian | 0.56 | 7.13 | 7.85 | 2805.92 | 0.86 |
| Severe land degradation | 55 | Gaussian | 0.46 | 5.93 | 7.76 | 2769.55 | 0.95 |
| Extreme land degradation | 57 | Gaussian | 0.01 | 3.01 | 0.33 | 2646.57 | 0.96 |

### 3.3. Correlation Analysis of Soil Organic Matter and Influencing Factors

According to the Pearson correlation coefficient (Table 4), the SOM values under four different land degradation conditions have a significant positive correlation with altitude, aspect, surface temperature and soil nutrients (i.e., total nitrogen, total phosphorus and total potassium) ($p < 0.01$), and negative correlation with slope, bulk density and pH ($p < 0.01$). This implies that both the surface temperature and soil nutrient content rises with increasing SOM. For mild and moderate land degradation, there was a significant positive correlation between elevation and soil organic matter ($p < 0.01$). There is a significant negative correlation ($p < 0.01$) between slope and the four types of land degradation, and a significant negative correlation ($p < 0.01$) between soil organic matter and bulk density and pH under extreme land degradation and severe land degradation conditions. The higher the compactness of soil, the stronger the acidity and alkalinity of soil, and the lower the SOM content. The correlation of the soil nutrients (i.e., total nitrogen, total phosphorus, total potassium) in different types of land degradation showed a significant positive correlation ($p < 0.01$).

**Table 4.** Correlations between SOM and influence factors for soil with different land degradation types.

| Soil Impact Factors | Types of Land Degradation | | | |
|---|---|---|---|---|
| | Mild Land Degradation | Moderate Land Degradation | Severe Land Degradation | Extreme Land Degradation |
| Altitude | 0.877 ** | 0.800 ** | 0.843 * | 0.781 * |
| Slope | −0.710 * | −0.739 ** | −0.737 ** | −0.793 ** |
| Aspect | 0.949 ** | 0.836 ** | 0.948 ** | 0.732 ** |
| Surface temperature | 0.800 ** | 0.915 ** | 0.930 ** | 0.821 ** |
| Bulk density | −0.689 * | −0.700 * | −0.952 ** | −0.841 ** |
| pH | −0.684 * | −0.890 * | −0.758 ** | −0.774 ** |
| Total nitrogen | 0.694 ** | 0.731 ** | 0.864 ** | 0.836 ** |
| Total phosphorus | 0.844 ** | 0.780 ** | 0.852 ** | 0.861 ** |
| Total potassium | 0.714 ** | 0.711 ** | 0.873 ** | 0.796 ** |

* The correlation was significant at the 0.05 level. ** The correlation was significant at the 0.01 level.

### 3.4. Spatial Distribution Pattern of SOM

The overall SOM content follows a downward trend with increasing land degradation, which is consistent with previous research results. From the perspective of different degrees of land degradation (Figure 2), most of the high-value areas of SOM are distributed in the areas without obvious degradation, mild degradation and moderate degradation. The distribution of land degradation in the study area was delineated based on the combination of Quickbird images with a resolution of 0.5 m, pictures captured by unmanned aerial vehicle (UAV) and handheld GPS. According to Figures 3 and 4, the SOM content in some areas with a high degree of land degradation is also high, which is consistent with the results shown in Table 4. The main reasons for this are as follows: (1) as the altitude increases, the temperature decreases, encouraging SOM accumulation; (2) different altitudes lead to distinct degrees of land degradation, which indirectly causes SOM differences; (3) in the low mountain and hilly areas, the larger slope and soil density, loose soil and low vegetation coverage lead to heavier SOM loss. There were significant differences between severe and extreme land degradation types. Combined with Table 2, it can be seen that the average levels of organic matter under severe land degradation and extreme land degradation are the lowest (18.36 and 15.39 g/kg), which can be explained by two aspects: on the one hand, the heavy and extreme land degradation weakens the soil surface aggregation and narrows the fluctuation range, which is also shown by the maximum and minimum values; on the other hand, the fewer samples lead to the greater randomness of content distribution.

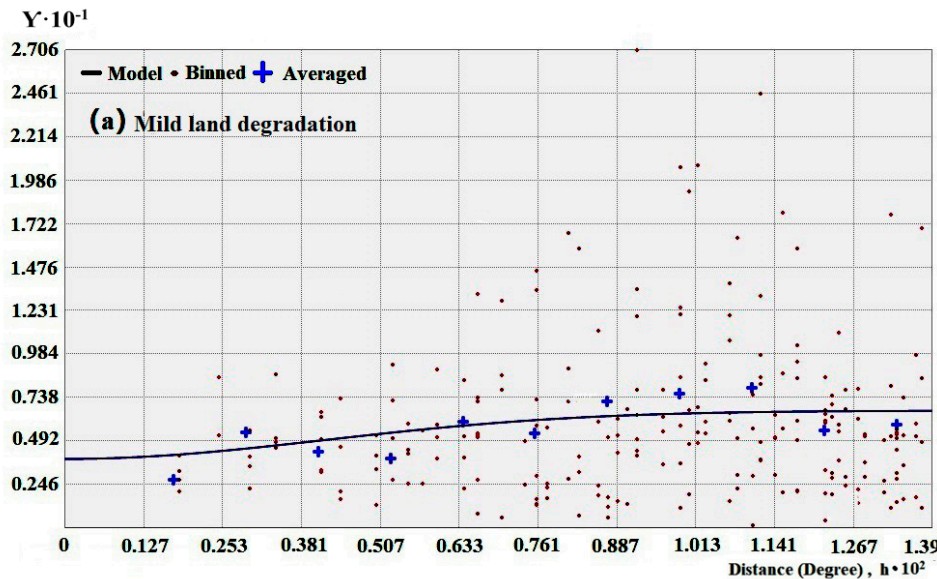

**Figure 2.** *Cont.*

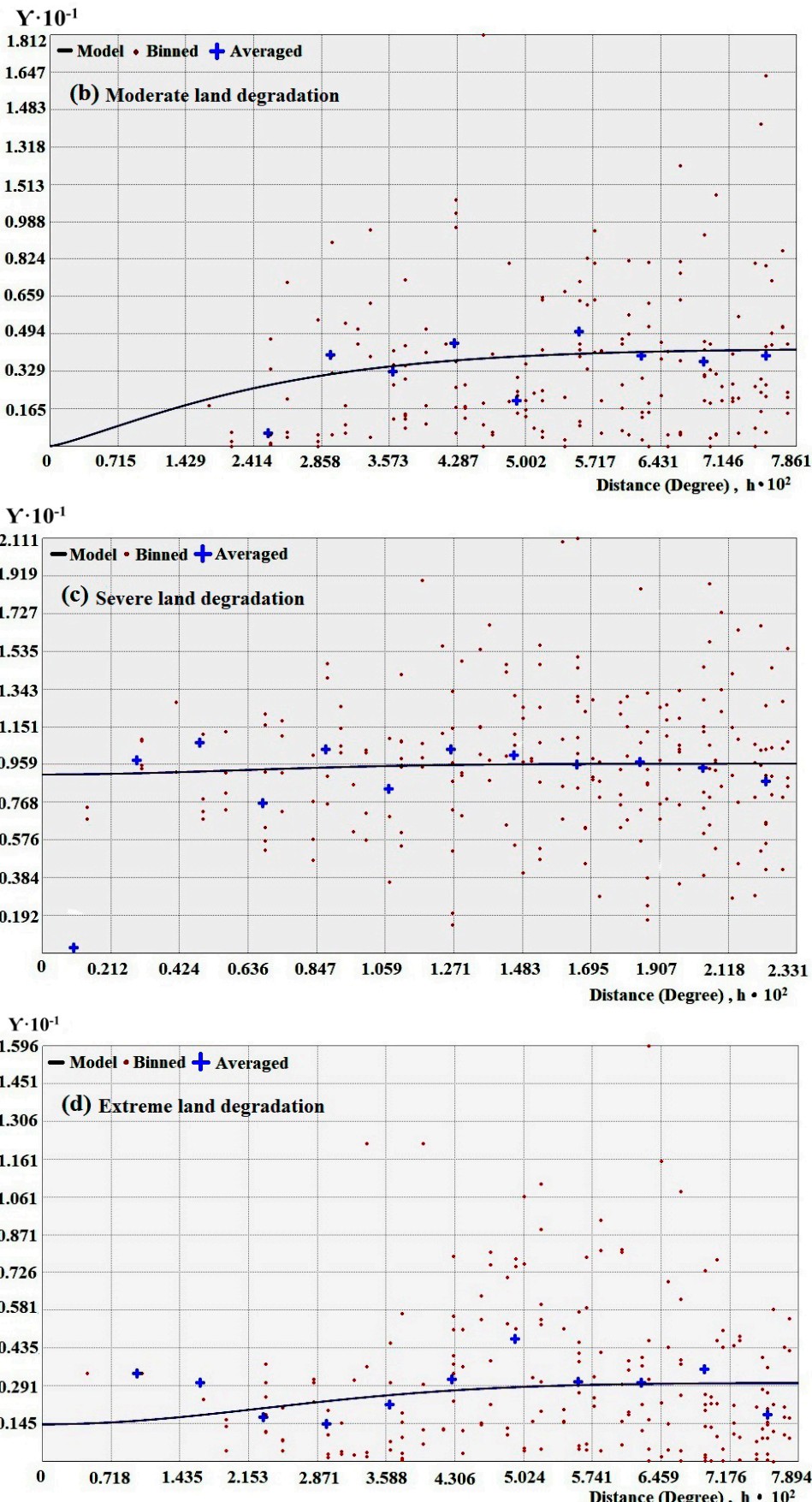

**Figure 2.** Experimental semi-variograms and fitted models for SOM different land degradation types.

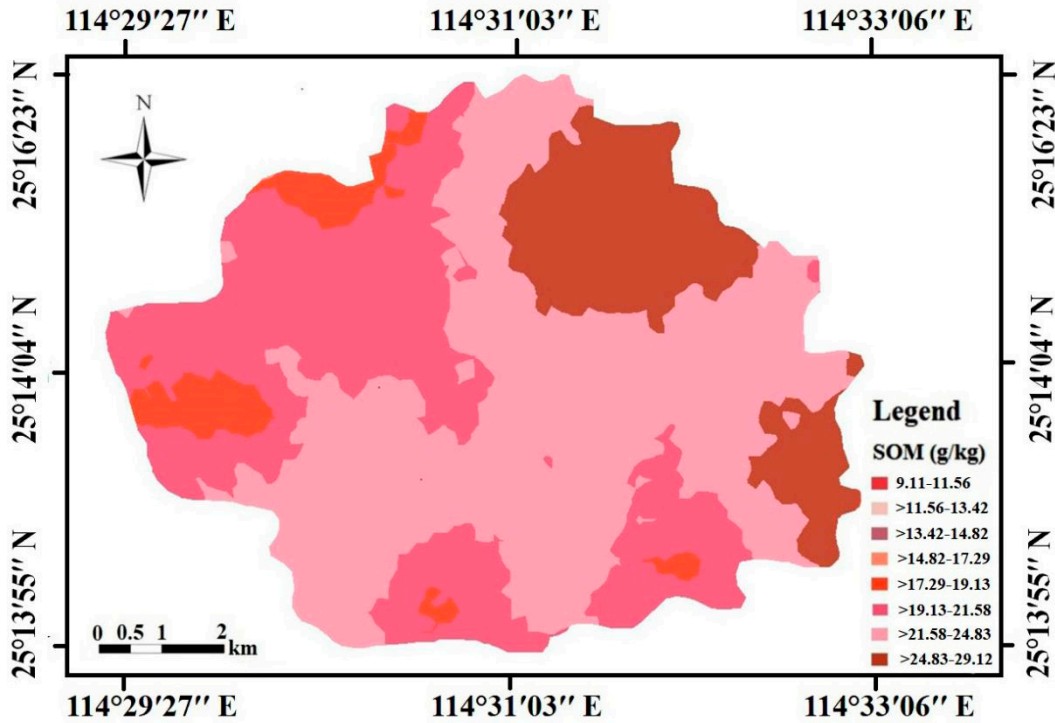

**Figure 3.** Spatial distribution of SOM.

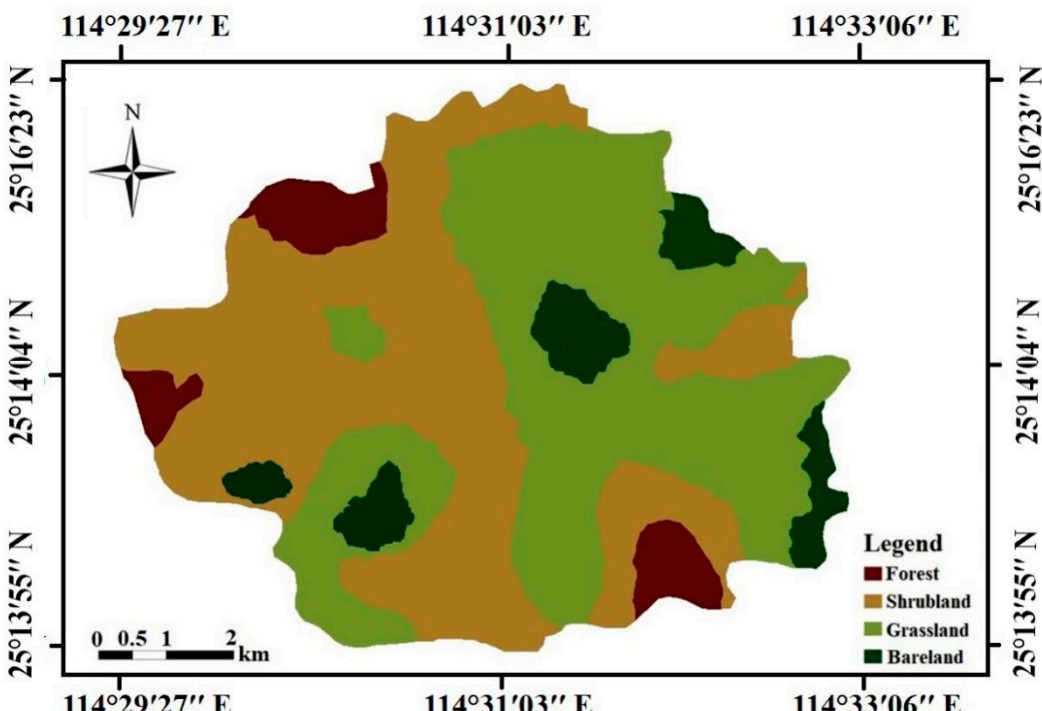

**Figure 4.** Distribution of different types of land degradation.

## 4. Discussion

Through the semivariogram analysis of SOM, the nugget effect value decreased with the aggravation of land degradation, indicating that the influence of terrain factors, and physical and chemical soil factors on SOM increased during the process of land degradation. Secondly, after the severe land degradation reached the minimum, the slope

had a significant negative correlation with the organic matter content. The nugget value decreased with the increase in land degradation, and there was a strong correlation between SOM content, topographic factors, and soil physical and chemical factors (Table 4). It can be observed that different factors influence the spatial variability of SOM under different types of land degradation. Therefore, land degradation is recognized as a major environmental problem that adversely depletes SOM, which in turn directly affects soils, their fertility, productivity and overall quality [28–30]. Liu et al. [19] also obtained similar results in studying the spatial variation characteristics of SOM in the Loess Plateau. He believed that external factors such as precipitation, temperature, and elevation had significant effects on the distribution of SOM.

Due to the influence of topography, parent material, climate, and biological and human activities, the spatial variability of SOM varies under different land degradation types (Table 2), and the influencing factors are also different (Table 4). Under mild and moderate land degradation types, elevation and aspect are the main factors affecting the spatial variation of SOM. It can be seen from Table 4 that the SOM content under different land degradation types varies significantly with altitude, showing an increasing trend with the increase in altitude. This is because, in low altitude areas, there are more human activities, a high intensity of land development and utilization, and a low content of SOM. Although the SOM values in some intensive cultivation areas are high, the scope is small. As the altitude rises, human farming activities decrease, and a large number of terraces were built in this area [31]. The higher degree of terracing also reduces soil erosion, avoiding soil erosion of cultivated land and maintaining the content of SOM. Under extreme and severe land degradation conditions, the dominant factors are slope and soil nutrients (total N, total P, total K). A large number of studies [32–34] have shown that the loss of nutrient elements in the surface soil caused by soil erosion on sloping farmland is the main reason for the decline in soil fertility. With the increase in slope and precipitation runoff, the rates of soil erosion, soil and water loss, soil nutrient loss and SOM loss accelerate.

Land degradation is characterized by spatial and temporal scales [35]. The temporal scale can be categorized into small areas, slopes, small watersheds and regions, which have distinct dominant and controlling processes. It is an effective measure to prevent soil erosion and land degradation by constructing terraces under different slopes to optimize land uses. The loss of SOM and other soil nutrients caused by land degradation destroys land resources, reduces land productivity, and aggravates floods and droughts. The increasingly serious land degradation poses a great threat to cultivated land use, soil and water conservation, and has become one of the major global environmental problems [36–38]. The Nanxiong basin is an ecologically fragile red bed area in South China with complex topography [16–18,31]. If the vegetation and soil layer on the surface of the red bed soil is destroyed, the soft rock of the base red bed will be rapidly weathered and form surface clasts. The precipitation runoff will remove the surface clasts continuously [39]. Gully erosion occurs after the surface is exposed, and new soil layers are difficult to regenerate. With the aggravation and degradation of land, SOM and nutrients are rapidly lost, and desertification occurs eventually with bare bedrock [40–42]. In recent decades, a lot of measures have been carried out to prevent and control the soil erosion of cultivated land, and remarkable results have been achieved, though serious problems of soil erosion still exist [17,31]. The spatial distribution of SOM under different degrees of land degradation is closely related to other factors, such as vegetation coverage, investment in soil and water conservation projects, land use, rainfall, and so on.

## 5. Conclusions

The results show that the SOM content is relatively low under different land degradation types in the ecologically vulnerable red bed areas of South China, and the SOM content tends to decrease with the exacerbation of land degradation. The overall variation degree of SOM in the study area is moderate, and the nugget effect values of SOM under different land degradation types are less than 25%, suggesting a strong spatial correlation.

The impact of terrain and soil structural parameters on the spatial variation of SOM reaches 89%. Under mild and moderate land degradation types, the elevation and aspect were the main parameters affecting the spatial variability of SOM. For extreme and severe land degradation types, slope and soil nutrients (i.e., total nitrogen, total phosphorus and total potassium) were the main factors. Compared with the previous research, the results of this study provide more direct guidance for investigations of the small-scale spatial variability of SOM in red bed areas. This research also paves the way for the future study of the large-scale spatial distribution of SOM content. In the future, the effects of biological and environmental factors, such as vegetation, soil, topography, geology and human activities, could be considered comprehensively to study the SOM impact mechanism. It is worth mentioning that regional differences exist for the widely distributed red beds around the world. The study area in this study has a subtropical monsoon climate, and the results can only represent the spatial distribution of SOM in the typical subtropical humid area of South China. More comprehensive studies for red bed areas in other climate zones could be conducted in the future to better understand the topic.

**Author Contributions:** Ping Yan and Kairong Lin conceived the original ideas. Kairong Lin, Yiren Wang and Xinjun Tu acquired the funding support. Ping Yan and Yiren Wang designed and performed the experiments. Kairong Lin, Ping Yan and Chunmei Bai outlined the paper. Ping Yan wrote the paper under guidance from Kairong Lin, and Ping Yan and Yiren Wang collected soil samples in the field. Chunmei Bai and Luobin Yan improved the paper and corrected English. All authors have read and agreed to the published version of the manuscript.

**Funding:** This work was supported by the National Natural Science Foundation of China (No. 51779279, No. 51822908, No. 41771008 and 41901005), the Fundamental Research Funds for the Central Universities of China (No. 20lgpy158, No. 19lgpy257), the Project funded by China Postdoctoral Science Foundation (No. 2020M672941 and No. 2020M672970), the Fundamental Research Funds for the Central Universities (SWU 118202) and Guangdong Provincial Natural Science Foundation Project (No. 2020A1515010438).

**Institutional Review Board Statement:** Not applicable.

**Informed Consent Statement:** Not applicable.

**Data Availability Statement:** The data presented in this study are available on request from the corresponding author.

**Conflicts of Interest:** The authors declare no conflict of interest.

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
