# Peer review of "Assessment of Influencing Factors on the Spatial Variability of SOM in the Red Beds of the Nanxiong Basin of China, Using GIS and Geo-Statistical Methods"

_ijgi, doi:10.3390/ijgi10060366_

Round 1

Reviewer 1 Report

1) L16-32: The abstract should be a total of about 200 words maximum. Describe briefly the main methods or treatments applied.

2) L33: Please find such words which are not in the title.

3) L36-82: Please briefly mention the main aim of the work and highlight the main novelty of this paper.

4) L110-113: Section 2.3 is too short. Please combine this section with another. 

5) L146: Table 2 is in the wrong place.

6) The % sign should be after the number, no space before the '%'.

7) L184: Figure 2 is hardly legible.

8) L222: Figure 3 and 4 are hardly legible, the legend is illegible. Maybe combine them into one figure?

9) In Discussion section, the authors should discuss the results of the research and compare them with other studies. In the discussion should be cited appropriate literature.

10) L275: In Conclusions section, the authors should recommend what areas show the most promise or what need more research and provide the weaknesses and limitation of the research.

Author Response

Comments and Suggestions for Authors

1) L16-32: The abstract should be a total of about 200 words maximum. Describe briefly the main methods or treatments applied.

Response: Thanks for pointing out this problem. The text has been revised and the text now reads:  Soil organic matter (SOM) plays an important role in improving soil fertility and crop productivity. Studying its spatial variability can provide scientific basis for precision fertilization and land degradation control. In this study, geostatistics and classical statistical methods were used to analyze the spatial variability of SOM, and to explore the spatial variability of SOM and its influencing factors under different degrees of land degradation in the red bed area of southern China. The results showed that: with the deepening of land degradation, the content of SOM showed a downward trend. There were significant differences in SOM content among different land degradation degrees. The coefficient of variation from large to small was as follows: extreme land degradation (13.61%) > mild land degradation (8.98%) > moderate land degradation (7.96%) > severe land degradation (5.64%). Under mild and moderate land degradation conditions, there was a significant positive correlation between altitude and SOM (P < 0.01). Bulk density and pH value were significantly negatively correlated with SOM (P < 0.01). It can be seen that terrain factors and soil physical and chemical factors have great influence on SOM.

2) L33: Please find such words which are not in the title.

Response: Thanks for pointing out this problem. Corrections have been done. The keywords have been revised as follows: soil organic matter; land degradation; semivariogram; spatial distribution; classical statistics.

3) L36-82: Please briefly mention the main aim of the work and highlight the main novelty of this paper.

Response: Thanks for pointing out this problem. The text has been revised and the text now reads: 

“Therefore, the main objectives of this study are: (1) to clarify the spatial variation characteristics of SOM under different land degradation types in this study area; (2) to obtain the influencing factors of spatial variation of SOM through correlation analysis; (3) to explore the impact mechanism of land degradation on the spatial distribution pattern of SOM.”

4) L110-113: Section 2.3 is too short. Please combine this section with another.

Response: Thanks for pointing out this problem. The supplement is in Section 2.3 Laboratory testing.

5) L146: Table 2 is in the wrong place.

Response: Thanks for pointing out this problem. Corrections have been done.

According to the statistical analysis results of SOM (Table 2).

6) The % sign should be after the number, no space before the '%'.

Response: Thanks for pointing out this problem. Corrections have been done.

7) L184: Figure 2 is hardly legible.

Response: Thanks for pointing out this problem. Corrections have been done.

8) L222: Figure 3 and 4 are hardly legible, the legend is illegible. Maybe combine them into one figure?

Response: Thanks for pointing out this problem. Corrections have been done.

9) In Discussion section, the authors should discuss the results of the research and compare them with other studies. In the discussion should be cited appropriate literature.

Response: Thanks for pointing out this problem. The text has been revised and the text now reads: “Liu et al. [19] also got similar results in studying the spatial variation characteristics of SOM in different land uses in the Loess Plateau. He believed that external factors such as precipitation, temperature, elevation had significant effects on the distribution of SOM”

10) L275: In Conclusions section, the authors should recommend what areas show the most promise or what need more research and provide the weaknesses and limitation of the research.

Response: Thanks for pointing out this problem. The text has been added and the text now reads: Finally, due to the wide distribution of red beds in the world, the red beds in different regions have certain regional differences. The study area belongs to the subtropical monsoon red beds area, and the spatial variability of SOM can only represent the spatial variability of red beds SOM in the typical subtropical humid area of South China. It is difficult to reflect the spatial variability of SOM in other temperature zones. Therefore, it is necessary to increase the comparative study of different zones in the future.

Reviewer 2 Report

This is a well-written, well-organized and well-illustrated paper. It presents the results of original research and makes a valuable contribution to knowledge and understanding of spatial variability of soil organic matter based on GIS and Geo-Statistical Methods.

Specific comments in the attachment. 

Author Response

Review 2

Comments and Suggestions for Authors

This is a well-written, well-organized and well-illustrated paper. It presents the results of original research and makes a valuable contribution to knowledge and understanding of spatial variability of soil organic matter based on GIS and Geo-Statistical Methods.

Specific comments in the attachment.

Please accept after minor revision.

Specific comments:

  1. Figure 3 and 4, please add coordinates.

Response: Thanks for pointing out this problem. Corrections have been done.

  1. Try to provide color figure in 1. (Elevation figure instead of gray).

Response: Thanks for pointing out this problem. Corrections have been done.

  1. Section 2.3, How many samples were taken?

Response: Thanks for pointing out this problem. The text has been revised and the text now reads: “According to the different sampling density and considering the soil type, the random grid was set up under four types of land degradation (mild, moderate, severe and extreme), and a total of 225 sample points (typical fields) were set up.”

  1. Try to provide sample location in the figure. (New figure) or provide table with

location name and coordinates.

Response: Thanks for pointing out this problem. Corrections have been done.

  1. Is it possible to provide laboratory name?

Response: Thanks for pointing out this problem. Laboratory name is Key Laboratory of comprehensive management of agricultural environment of Guangdong Province, Institute of eco-environmental and soil science, Guangdong academy of sciences, China.

  1. Need more explanation in the section 2.3.

Response: Thanks for pointing out this problem. The supplement is in Section 2.3 Laboratory testing.

Reviewer 3 Report

The manuscript has made an attempt to explore the relationship between SOM and other related factors with land degradation, from a statistical perspective. It is quite interesting. However, the authors need to keep in mind that certain things can not be concluded based purely on the statistical outcomes, as SOM and land degradation are natural systems that may behave in very different ways under different land management techniques, even with similar 'numbers'. Hence it is needed to put the result into context, and then arrive at a 'conclusion'.

Title: The parameters - physical and chemical - have been pre-determined as I understand. Hence, using the phrase 'Influencing factors ... based on GIS and Geo-statistical methods...' is somewhat misleading. Maybe "Assessment of Influencing Factors on the Spatial Variability of SOM in the Red Beds of the Nanxiong Basin of China, Using GIS and Geo-Statistical Methods" or "Assessing the Factors Influencing Spatial Variability of SOM in the Red Beds of the Nanxiong Basin of China, Using GIS and Geo-Statistical Methods". These are merely suggestions, for the authors to think.

Abstract: line 5 - '... deepening of land degradation ...' would read better as '... increase in land degradation ...' 

Words that seem out of context - eg. line 68 'silent', line 256 'returning'

Line 67 - give more references for 'multiple studies'

Lines 76 - 82 - it is expected to observe very low levels of SOM in a degraded land. However, the influencing factors for the loss of SOM would normally differ from place to place. Hence, it would read better if the authors reminded the reader that the set objective is focused to this particular landscape, but is having a high potential to be applied to a very similar area/environment.

Line 92 - are the heights above the MSL or some other datum? This has not been specified.

Line 113 - the sentence is incomplete. More information on the laboratory technique is needed to replicate the work by another, if needed.

Line 125 onwards - the described 'land degradation classification' system, is this a system that has been used previously by any researcher? If so, where is/are the references? If not, how did the authors develop the classification system? Is the information presented in Table 1 developed by the authors or extracted? How were the detailed information developed? How was the land delineated? More detailed information is needed under this sub-section.

Line 146 - Table 1 - where there is no A horizon to a soil, it is generally understood that there is no SOM. In that case, how did these categories end up with a certain value for SOM? It is not impossible, but how would you explain it?

Line 146 - Table 2 discusses about 225 samples. What are these samples and how have they been collected? 

How did you obtain the SOM? Did you calculate it using certain equations, or did you measure it directly? It has not been clearly mentioned.

Line 167 - the unit of measure of SOM is 'm'. How is this possible?

Lines 188 - 189 - it is implied that SOM increases with increasing surface temperature and soil nutrient. It happens the other way - both the soil temperature and nutrient levels increase due to SOM.

Line 190, 226 - 228 - the SOM is shown to be negatively correlated with the slope in this particular study. But there are many cases where there can be no SOM loss on steep slopes, provided that the land management techniques are sustainable. Therefore it is not correct to imply that it is natural for SOM to decrease with an increasing slope angle.

Line 196 - 197 - how is it possible to show both acidity and alkalinity simultaneously? You can have only one at a given time.

Line 197 - 199 - it is of course obvious that the soil nutrients will be lost with land degradation. That does not tell the reader anything new, simply shows that the statistics are pointing out the obvious. It is the new finding that you need to point out here.

Line 226 - The sentence '... severe land degradation reached the minimum ...' is not very clear. Please phrase it more clearly.

Line 236 - Table 2 does not show the information that it is said to show in the text. Or, are you saying that Table 2 shows the land degradation types? Please phrase it clearly.

Line 242 - not 'generally', but as a consequence.

Line 250 - 252 - this is not the case at all times. Therefore, this statement needs to be put in context.

Line 272 - not 'in', but 'under'.

Line 276 - 279 - this is obvious and expected. What are the new findings from this particular research? 

Author Response

Review 3

Comments and Suggestions for Authors

The manuscript has made an attempt to explore the relationship between SOM and other related factors with land degradation, from a statistical perspective. It is quite interesting. However, the authors need to keep in mind that certain things can not be concluded based purely on the statistical outcomes, as SOM and land degradation are natural systems that may behave in very different ways under different land management techniques, even with similar 'numbers'. Hence it is needed to put the result into context, and then arrive at a 'conclusion'.

(1)Title: The parameters - physical and chemical - have been pre-determined as I understand. Hence, using the phrase 'Influencing factors ... based on GIS and Geo-statistical methods...' is somewhat misleading. Maybe "Assessment of Influencing Factors on the Spatial Variability of SOM in the Red Beds of the Nanxiong Basin of China, Using GIS and Geo-Statistical Methods" or "Assessing the Factors Influencing Spatial Variability of SOM in the Red Beds of the Nanxiong Basin of China, Using GIS and Geo-Statistical Methods". These are merely suggestions, for the authors to think.

Response: Thanks for pointing out this problem. We have revised the title to "Assessment of Influencing Factors on the Spatial Variability of SOM in the Red Beds of the Nanxiong Basin of China, Using GIS and Geo-Statistical Methods".

(2)Abstract: line 5 - '... deepening of land degradation ...' would read better as '... increase in land degradation ...'

Response: Thanks for pointing out this problem. Corrections have been done.

(3)Words that seem out of context - eg. line 68 'silent', line 256 'returning'

Response: Thanks for pointing out this problem. line 68: “Silent content” has been deleted. line 256 'returning', the associated sentence has been changed to It is an effective measure to prevent soil erosion and land degradation by constructing terraces under different slopes to optimize the land use pattern.

(4)Line 67 - give more references for 'multiple studies'

  1. Response:Thanks for pointing out this problem. Corrections have been done. Relevant references [19-23] have been added. 
  2. Liu, Z., Shao, M., Wang, Y. Effect of environmental factors on regional soil organic carbon stocks across the Loess Plateau region, China. Agriculture ecosystems and environment 2011, 142, 184-194.
  3. Tonitto, C. , Goodale, C. L. , Weiss, M. S. , Frey, S. D. , Ollinger, S. V. The effect of nitrogen addition on soil organic matter dynamics: a model analysis of the harvard forest chronic nitrogen amendment study and soil carbon response to anthropogenic n deposition. Biogeochemistry2014,117(2-3), 431-454. 1
  4. Yones, K., Farshad, K., Sohaila, E. The effect of land use change on soil and water quality in northern iran. Journal of Mountain Science2012, 6, 74-92.
  5. Meersmans, J., Ridder, F. D., Canters, F., Baets, S. D., Molle, M.V. A multiple regression approach to assess the spatial distribution of soil organic carbon (soc) at the regional scale (flanders, belgium). Geoderma, 2008, 143(1-2), 1-13.
  6. Hu, P.L., Liu, S.J., Ye, Y.Y., Wei, Z., Su, Y.R. Effects of environmental factors on soil organic carbon under natural or managed vegetation restoration. Land Degradation and Development,2018, 29, 387-398.

(5)Lines 76 - 82 - it is expected to observe very low levels of SOM in a degraded land. However, the influencing factors for the loss of SOM would normally differ from place to place. Hence, it would read better if the authors reminded the reader that the set objective is focused to this particular landscape, but is having a high potential to be applied to a very similar area/environment.

Response: Thanks for pointing out this problem. The text has been revised and the text now reads: In view of this, this study takes the soil organic matter in the typical ecological fragile red bed area of China as the research object, aiming at the special lithology of the red bed, and discusses the relationship between the soil organic matter content of different types of land degradation and topographic factors, soil physical and chemical factors.Therefore, the main objectives of this study are: (1) to clarify the spatial variation characteristics of SOM under different land degradation types in this study area; (2) to obtain the influencing factors of spatial variation of SOM through correlation analysis; (3) to explore the impact mechanism of land degradation on the spatial distribution pattern of SOM in this study area.

(6)Line 92 - are the heights above the MSL or some other datum? This has not been specified.

Response: Thanks for pointing out this problem. In order not to affect readers' misunderstanding, we have deleted “Line 92-a relative altitude of less than 50 m”.

(7)Line 113 - the sentence is incomplete. More information on the laboratory technique is needed to replicate the work by another, if needed.

Response: Thanks for pointing out this problem. The supplement is in Section 2.3 Laboratory testing.

(8)Line 125 onwards - the described 'land degradation classification' system, is this a system that has been used previously by any researcher? If so, where is/are the references? If not, how did the authors develop the classification system? Is the information presented in Table 1 developed by the authors or extracted? How were the detailed information developed? How was the land delineated? More detailed information is needed under this sub-section.

Response: Thanks for pointing out this problem. The “Land degradation classification”system”was developed by one of the authors Luobin Yan. This system was developed mainly through a lot of field work. These detailed information presented was developed according to plot investigation in Nanxiong Basin. The distribution of different types of land degradation of study area was delineated based on the combination of Quickbird image with resolution of 0.5m, pictures captured by UAV (Unmanned Aerial Vehicle) and Handheld GPS.

(9)Line 146 - Table 1 - where there is no A horizon to a soil, it is generally understood that there is no SOM. In that case, how did these categories end up with a certain value for SOM? It is not impossible, but how would you explain it?

Response: Sorry for causing confusion. As we stated in section 2.4, soil is not totally depleted in area with Extreme land degradation rather than sporadically distributed as small patches with thickness less than 5cm. Actually, we collected samples from soil layer rather than rock fragments.

(10)Line 146 - Table 2 discusses about 225 samples. What are these samples and how have they been collected?

Response: Thanks for pointing out this problem. The relevant information of soil collection is described in detail in Section 2.2.

(11)How did you obtain the SOM? Did you calculate it using certain equations, or did you measure it directly? It has not been clearly mentioned.

Response: Thanks for pointing out this problem. The supplement is in Section 2.3 Laboratory testing.

(12)Line 167 - the unit of measure of SOM is 'm'. How is this possible?

Response: Thanks for pointing out this problem. 'm 'is the unit of SOM range.

(13)Lines 188 - 189 - it is implied that SOM increases with increasing surface temperature and soil nutrient. It happens the other way - both the soil temperature and nutrient levels increase due to SOM.

Response: Thanks for pointing out this problem. Corrections have been done. It is implied that SOM increases with increasing surface temperature and soil nutrient.

(14)Line 190, 226 - 228 - the SOM is shown to be negatively correlated with the slope in this particular study. But there are many cases where there can be no SOM loss on steep slopes, provided that the land management techniques are sustainable. Therefore it is not correct to imply that it is natural for SOM to decrease with an increasing slope angle.

Response: Thanks for pointing out this problem. Yes, we also agree with you that the negative correlation between SOM and slope obtained in this study is only a result based on the experiment of this study, which is not universal. Therefore, we have deleted the conclusion of judgment (Line 226 - 228).

(15)Line 196 - 197 - how is it possible to show both acidity and alkalinity simultaneously? You can have only one at a given time.

Response: Thanks for pointing out this problem. The text has been revised and the text now reads: The higher the compactness of soil, the stronger the acidity of soil, and the lower the content of organic matter.

(16)Line 197 - 199 - it is of course obvious that the soil nutrients will be lost with land degradation. That does not tell the reader anything new, simply shows that the statistics are pointing out the obvious. It is the new finding that you need to point out here.

Response: Thanks for pointing out this problem. The text has been added and the text now reads: It can be seen that there is a significant correlation between SOM and topographic factors in the red bed ecological vulnerable area, which have important practical significance for guiding the soil improvement in the red bed ecological vulnerable area.

(17)Line 226 - The sentence '... severe land degradation reached the minimum ...' is not very clear. Please phrase it more clearly.

Response: Thanks for pointing out this problem. The text has been revised and the text now reads: Secondly, in moderate, severe and extreme degree of land degradation, slope and SOM content were significantly negatively correlated, and the correlation coefficients were -0.739, - 0.737, - 0.793, respectively. The correlation of severe land degradation was the largest, indicating that slope played a major role in the spatial variation of SOM in severe land degradation.

(18)Line 236 - Table 2 does not show the information that it is said to show in the text. Or, are you saying that Table 2 shows the land degradation types? Please phrase it clearly.

Response: Thanks for pointing out this problem. The correct one should be table 3, which we have revised

(19)Line 242 - not 'generally', but as a consequence.

Response: Thanks for pointing out this problem. Corrections have been done. We have deleted 'generally'.

(20)Line 250 - 252 - this is not the case at all times. Therefore, this statement needs to be put in context.

Response: Thanks for pointing out this problem. We have put this statement in part 3.3 context.

(21)Line 272 - not 'in', but 'under'.

Response: Thanks for pointing out this problem. Corrections have been done.

(22)Line 276 - 279 - this is obvious and expected. What are the new findings from this particular research?

Response:Thanks for pointing out this problem. The text has been added and the text now reads:

It can be seen that the loss of SOM plays an important role in the process of red bed land degradation. It is of great scientific significance to objectively reveal the mechanism of red bed desertification for promoting the red bed land degradation control and sustainable development in the humid region of southern China.

Reviewer 4 Report

Dear Authors,

After careful reading my decision is: accept in present form.

Sincerely yours,

Reviewer

Author Response

We greatly appreciate the reviewer for the thoughtful and encouraging comments on our manuscript.

Round 2

Reviewer 1 Report

In my opinion, this paper is interesting. The manuscript has gone through a significant revision as compared to the earlier version. But, the authors did not successfully address to my previous comments. In particular to my previous comment no 9 (about discussion section).
Although the authors provide one paragraph in response to my comment, I do not find it in the revised manuscript. Moreover, there is still too little references in the discussion. 
In Discussion section, the authors should discuss the results of the research and compare them with other studies. In the discussion should be cited appropriate literature.
This should be changed and completed. I encourage the authors to make greater effort.

Author Response

Thanks for pointing out this problem. The text has been revised and the text now reads: “Therefore, land degradation is recognized as a main environmental problem that adversely depletes SOM, which in turn directly affects soils, their fertility, productivity and overall quality[28-30]. Liu et al.[19] also got similar results in studying the spatial variation characteristics of SOM in the Loess Plateau. He believed that external factors such as precipitation, temperature, elevation had significant effects on the distribution of SOM. ” The related literature is as follows:

28.Phesheya, D., Pauline, C., Alan, M., Vincent, C. Land degradation impact on soil organic carbon and nitrogen stocks of sub-tropical humid grasslands in south africa - sciencedirect. Geoderma 2014, 235–236, 372-381.

29.Martinsen, V.,Mulder, J., Austrheim, G.,Mysterud, A. Carbon storage in low-alpine grassland soils: effects of different grazing intensities of sheep. Eur. J. Soil Sci. 2011, 62, 822–833.

30.Steffens, M.,Kölbl, A., Totsche, K.U.,Kögel-Knabner, I. Grazing effects on soil chemical and physical properties in a semiarid steppe of Inner Mongolia (P.R. China). Geoderma 2008, 143, 63–72.

31.Luo, G.S, Peng, H., Zhang, S.Y, Yan, L.B, Dong, Y.X. Exploring the variations of redbed badlands and their driving forces in the nanxiong basin, southern china: a geographically weighted regression with gridded data. Journal of Sensors 2021, 8, 1-13.

35.Negrete-Yankelevich, S., Porter-Bolland, L., Blanco-Rosas, J.L., Barois, I. Historical roots of the spatial, temporal, and diversity scales of agricultural decision-making in sierra de santa marta, los tuxtlas. Environmental Management 2013, 52, 45-60.

36.Chamizo, S., Cantón Y., Rodríguez-Caballero, E., Domingo, F., Escudero, A. Runoff at contrasting scales in a semiarid ecosystem: a complex balance between biological soil crust features and rainfall characteristics. Journal of Hydrology 2012, 452-453, 130-138.

37.Wessels, K.J., Prince, S.D., Malherbe, J., Small, J., Frost, P.E., Vanzyl,D. Can human-induced land degradation be distinguished from the effects of rainfall variability? a case study in south africa. Journal of Arid Environments 2007, 68(2), 271-297.

38.Dregne, H.E., Kassas, M., Rozanov, B. A new assessment of the world status of desertification. Desertification Control Bulletin 1991, 20, 6–19.

39.Justice, C.O., Dugdale, G., Townshend, J.R.G., Narracott, A.S., Kumar, M. Synergism between NOAAAVHRR and Meteosat data for studying vegetation development in semi-arid West Africa. International Journal of Remote Sensing 1991, 12, 1349–1368.

40.Clarke, M.L, Rendell, H.M. Process–form relationships in southern italian badlands: erosion rates and implications for landform evolution. Earth Surface Processes and Landforms 2006, 31, 15-29.

41.Kevin, H., Marie-Franoise, A. New insights into rock weathering from high-frequency rock temperature data: an antarctic study of weathering by thermal stress. Geomorphology 2001, 41, 23-35.

42.Nadal-Romero, E., Martinez-Murillo, J.F. , Vanmaercke, M., Poesen, J. Scale-dependency of sediment yield from badland areas in mediterranean environments. Progress in Physical Geography 2011, 35, 297-332.

Reviewer 3 Report

The authors have addressed all comments and queries raised by me. As such, I am happy to accept it in the present form.

Slight modification though, to line 226. The correct wording is;

It implies that both the surface temperature and soil nutrient content rises with increasing SOM.

Author Response

Response: Thanks for pointing out this problem. Corrections have been done.